# Drug Holidays and Overall Survival of Patients with Metastatic Colorectal Cancer

**DOI:** 10.3390/cancers13143504

**Published:** 2021-07-13

**Authors:** Silvio Ken Garattini, Debora Basile, Marta Bonotto, Elena Ongaro, Luca Porcu, Carla Corvaja, Monica Cattaneo, Victoria Josephine Andreotti, Camilla Lisanti, Elisa Bertoli, Giacomo Pelizzari, Donatella Iacono, Gianmaria Miolo, Giovanni Gerardo Cardellino, Angela Buonadonna, Giuseppe Aprile, Gianpiero Fasola, Fabio Puglisi, Nicoletta Pella

**Affiliations:** 1Department of Oncology, ASUFC University Hospital of Udine, 33100 Udine, Italy; marta.bonotto@asufc.sanita.fvg.it (M.B.); giacomo.pelizzari@asufc.sanita.fvg.it (G.P.); donatella.iacono@asufc.sanita.fvg.it (D.I.); giovanni.cardellino@asufc.sanita.fvg.it (G.G.C.); gianpiero.fasola@asufc.sanita.fvg.it (G.F.); nicoletta.pella@asufc.sanita.fvg.it (N.P.); 2Department of Oncology, San Bortolo General Hospital, 36100 Vicenza, Italy; deborabasile1090@gmail.com (D.B.); giuseppe.aprile@aulss8.veneto.it (G.A.); 3Department of Medical Oncology, Unit of Medical Oncology and Cancer Prevention, Centro di Riferimento Oncologico di Aviano (CRO), IRCCS, 33081 Aviano, Italy; elena.ongaro@cro.it (E.O.); gmiolo@cro.it (G.M.); abuonadonna@cro.it (A.B.); fabio.puglisi@cro.it (F.P.); 4Laboratory of Methodology for Clinical Research, Department of Oncology, Istituto di Ricerche Farmacologiche Mario Negri IRCCS, 20156 Milan, Italy; luca.porcu@marionegri.it; 5Department of Medicine, University of Udine, 33100 Udine, Italy; carlacorvaja@gmail.com (C.C.); aprile83@gmail.com (M.C.); victoriaja888@gmail.com (V.J.A.); drlisanti.camilla@gmail.com (C.L.); bertoli.eli@gmail.com (E.B.)

**Keywords:** metastatic colorectal cancer, treatment strategies, drug holidays, maintenance

## Abstract

**Simple Summary:**

During first-line treatment of metastatic colorectal cancer, drug holidays (DHs) are usually adopted to limit toxicity. Literature lacks a formal demonstration that first-line continuous treatment, or maintenance, provides longer overall survival compared to DHs. We retrospectively studied the overall survival impact of DHs, demonstrating that a treatment break after initial induction chemotherapy may be considered in carefully clinically selected patients with metastatic colorectal cancer. Our study should reassure medical oncologists dedicated to colorectal cancer on the use of DHs.

**Abstract:**

Different de-escalation strategies have been proposed to limit the risk of cumulative toxicity and guarantee quality of life during the treatment trajectory of patients with metastatic colorectal cancer (mCRC). Programmed treatment interruptions, defined as drug holidays (DHs), have been implemented in clinical practice. We evaluated the association between DHs and overall survival (OS). This was a retrospective study, conducted at the University Hospital of Udine and the IRCCS CRO of Aviano. We retrieved records of 608 consecutive patients treated for mCRC from 1 January 2005 to 15 March 2017 and evaluated the impact of different de-escalation strategies (maintenance, DHs, or both) on OS through uni- and multivariate Cox regression analyses. We also looked at attrition rates across treatment lines according to the chosen strategy. In our study, 19.24% of patients received maintenance therapy, 16.12% DHs, and 9.87% both, while 32.07% continued full-intensity first-line treatment up to progression or death. In uni- and multivariate analyses first-line continuous treatment and early discontinuation (treatment for less than 3 months) were associated to worse OS compared to non-continuous strategies (HR, 1.68; 95% CI, 1.22–2.32; *p* = 0.002 and HR,4.89; 95% CI, 3.33–7.19; *p* < 0.001, respectively). Attrition rates were 22.8%, 20.61%, and 19.64% for maintenance, DHs, or both, respectively. For continuous therapy and for treatment of less than 3 months it was 21.57% and 49%. De-escalation strategies are safe and effective options. DHs after initial induction chemotherapy may be considered in clinically selected patients with metastatic colorectal cancer.

## 1. Introduction

Colorectal cancer (CRC) is the third-most-common and the fourth-most-lethal cancer worldwide [1]. In a recent projection, CRC-related deaths are expected to rise over the next 15 years. However, mortality rates are predicted to continue to decrease globally [2], due to early detection and improvement of surgical and loco-regional techniques in the metastatic setting [3]. Furthermore, the development of new chemotherapy combination regimens and the introduction of biologic agents in recent decades has also led to a further increase in survival [4,5,6,7,8,9]. Consequently, clinicians need to manage patient treatment for up to two to three years and to define sequences and durations of the treatments, avoiding a heavy treatment burden and controlling long term toxicities. Thus, various groups started to investigate milder therapeutic approaches introducing maintenances (interruption of a part of the upfront chemotherapy backbone), intermittent strategies (on/off treatment periods for preplanned times), and treatment-free drug holidays (DHs) in patients who have obtained disease control with induction combination chemotherapy.

Historically, discontinuation of oxaliplatin has been the first attempt to limit dose-related neurotoxicity, thus representing the paradigm of de-escalation to maintenance therapy. Different trials successfully evaluated oxaliplatin omission after doublet or triplet induction chemotherapy, with or without biologic agents, and “stop-and-go” approaches [10,11,12]. Maintenance therapy with fluoropyrimidines, combined or not to bevacizumab or anti-EGFR, may effectively reduce toxicities and improve patients’ quality of life without compromising clinical outcomes [13,14,15]. Unfortunately, maintenance with biological agents only [15,16,17,18] has not been successful so far. Maintenance treatment with a biological agent (i.e., bevacizumab) versus a drug holiday (DH) approach [19,20] led to interesting results. Several studies compared biologic drugs and/or fluoropyrimidine full-intensity (or maintenance therapy) with DHs, with contrasting results in terms of overall survival [20,21,22]. Moreover, the recent introduction of more intense chemotherapy regimens has led to higher response rates (RR) and more profound responses that generate longer progression-free survival (PFS) making clinicians more prone to offer DHs [23,24]. As no conclusive results on the overall survival (OS) benefit of continuous chemotherapy over maintenance or DHs have been achieved so far, the aim of this study is to evaluate the impact of de-escalation strategies (maintenance and, with special interest, DHs) on OS compared to continuous treatment. 

## 2. Results

### 2.1. Descriptive Analysis

The study included a cohort of 890 patients with a diagnosis of mCRC who received at least a first-line chemotherapy regimen. The final analysis was, however, performed on a population of 608 patients who had not received metastasectomies or loco-regional treatments within first-line treatment (clinical, pathological, and treatment characteristics are listed in Table 1). In the whole cohort, 64.14% of the patients were younger than 70 years. Of note, 28.95% had a right tumor location and 72.86% underwent surgery on the primary tumor. Approximately 40% had more than one metastatic site involved and the most frequent site of metastatic spread was the liver (33.55%), followed by the lungs (21.05%), peritoneum (20.23%), and lymph nodes (16.78%). A metastasectomy before first-line chemotherapy was performed in 21.05% of the whole cohort. As for the biological profile, KRAS, NRAS, and BRAF mutations were detected in 38.98%, 2.63%, and 7.40% of patients, respectively. First-line chemotherapy consisted of a combination of a triplet plus a biologic agent in 9.38% of cases, whereas 75.79% received a doublet-based regimen (approximately 44% doublet plus biologic agent and 31% doublet alone). After induction chemotherapy, treatment strategies included maintenance therapy in 19.24% of patients, a drug holiday in 16.12%, or both in 9.87%. Of note, 16.78% received first-line chemotherapy for less than 3 months overall. A second-line chemotherapy regimen was offered to 69.41% of the patients.

### 2.2. Survival Impact of De-Escalation Strategies

At the median follow-up of 70.32 months, median OS was 20.25 months. At univariate analysis for OS (Table 2), all-RAS mutations (HR, 1.38; 95% CI, 1.14–1.66; *p* = 0.001), ECOG performance status of 1 or 2 (1 vs. 0 HR, 1.99; 95% CI, 1.39–2.85; *p* = 0.000; 2 vs. 0 HR, 3.44; 95% CI, 2.18–5.44; *p* < 0.001, respectively), and number of metastatic sites > 1 (HR, 1.40; 95% CI, 1.16–1.67; *p* < 0.001) were associated with worse prognosis, as well as the extent of lymph nodes involvement, both for N2 (HR, 1.56; 95% CI, 1.19–2.05; *p* = 0.001) and N3 tumors (HR, 2.31; 95% CI, 1.61–3.30; *p* < 0.001). According to treatment execution after induction therapy, worse outcomes were observed for patients who had continuous combination chemotherapy without maintenance or drug holidays (HR, 1.57; 95% CI, 1.22–2.02; *p* = 0.001) and for those who had received chemotherapy for less than 3 months (HR, 3.75; 95% CI, 2.81–5.01; *p* < 0.001) (Figure 1). Median OS was 25.05, 29.79, and 30.25 months, respectively, in patients who had received maintenance therapy, drug holidays, or both (Figure 1). For patients who had continuous combination chemotherapy and those who had received chemotherapy for less than 3 months, median OS was 17.23 and 7.82 months, respectively. Conversely, left sidedness and rectum disease (HR, 0.66; 95% CI, 0.54–0.82; *p* < 0.001 and HR, 0.64; 95% CI, 0.51–0.81; *p* = 0.000, respectively), primary tumor resection (HR, 0.47; 95% CI, 0.39–0.58; *p* < 0.001) and metastasectomy before first-line chemotherapy (HR, 0.73; 95% CI, 0.59–0.91; *p* = 0.005) were associated with better OS (Table 2). Multivariate analysis for OS, left sidedness, and rectum disease confirmed to be independent prognostic factors of favorable survival outcomes (HR, 0.59; 95% CI, 0.45–0.79; *p* < 0.001 and HR, 0.56; 95% CI 0.41–0.77; *p* < 0.001, respectively). Moreover, the extent of nodal disease N2 (HR, 1.57; 95% CI, 1.14–2.17; *p* = 0.005) and a PS ECOG of 1 (HR, 2.48; 95% CI, 1.67–3.76; *p* < 0.001) confirmed independent unfavorable prognostic factors. Interestingly, a worse prognosis was confirmed for patients who had not been offered maintenance therapy or drug holidays and for those who were exposed to chemotherapy for less than 3 months (HR, 1.68; 95% CI, 1.22–2.32; *p* = 0.002 and HR, 4.89; 95% CI, 3.33–7.19; *p* < 0.001, respectively) (Table 2).

An ROC analysis was performed to develop a prognostic scoring model, based on selected factors that emerged in the multivariate analysis (every negative prognostic factor received a weighed score). The prognostic score aimed at identifying a threshold to discriminate patients with good and poor prognosis (Table 3). The cutoff identified was as score of 3 with higher than 3 predicting worse OS (Appendix A). Analyzing only the cohort of patients with a score <3 (good prognosis group), maintenance, holidays, or both did not show a significantly worse prognosis (HR, 0.75; 95% CI, 0.53–1.07; *p* = 0.112 and HR, 0.79; 95% CI, 0.53–1.17; *p* = 0.254, respectively). Obviously, the continuous chemotherapy (HR, 1.63; 95% CI, 1.22–2.19; *p* = 0.001) and a treatment duration less than 3 months (HR, 3.34; 95% CI, 2.36–4.73; *p* < 0.001) were associated with a worse outcome (Figure 2).

### 2.3. Attrition Rate across First-Line Treatment Strategies

In the overall population, the attrition rate (percentage of patients not achieving a further line of treatment) between first- and second-line chemotherapy was 27.62%. In particular, the attrition rate was 22.80%, 20.61%, and 19.64% for patients who had received maintenance therapy, drug holidays, or both, whereas attrition rates of 21.57% and 49%, respectively, were observed in the subgroups of patients who had continuous combination chemotherapy and those who had received chemotherapy for less than 3 months (*p* < 0.0001) (Figure 3). We looked for an association between de-potentiation strategies (DHs, maintenance, or both) and re-induction and we found it statistically significant (*p* = 0.012). The strongest association was between maintenance plus DHs and re-induction, followed by maintenance and then by DHs alone. 

## 3. Discussion

Advances in molecular biology and the introduction of novel therapeutic agents in mCRC treatment has led to significant improvement in survival, bringing the median OS up to about 30 months [25]. 

Though treatment de-escalation strategies are usually part of the treatment strategy used by medical oncologists in daily clinical practice, literature lacks a substantial formal demonstration of the efficacy of these approaches, especially if compared to continuous strategies. In a previous study, our group analyzed the use of DHs in a real world setting along with clinical and pathological factors associated with the choice of offering a DH [26].

The present study aimed to demonstrate the impact of de-escalation algorithms (with a special focus on DHs) on survival outcomes and whether these strategies resulted in detrimental survival in 608 consecutive mCRC patients. Maintenance treatment, drug holidays, maintenance followed by a break, and continuous treatment were analyzed in this study.

First, our real-world data confirmed that treatment de-escalation is a common practice. Indeed, 19.24%, 16.12%, and 9.87% of patients received maintenance, treatment holidays, or both, with the total percentage of de-escalated treatments reaching 45.23% in almost half of the cases.

Second, de-escalated treatment was associated with better OS compared to the continuous combination chemotherapy (HR, 1.68 for continuous treatment and HR, 4.89 for treatment <3 months), fostering shared decision-making of de-escalating strategies in daily clinical practice in carefully selected patients.

Evidence supporting de-escalating strategies is mostly derived from prospective studies. The UK MRCCR06 trial, the first assessing treatment holidays in patients with mCRC with stable or responding disease after 12 weeks of chemotherapy, detected fewer adverse events and no difference in terms of OS between continuous and intermittent treatment (HR, 0.87; 95% CI, 0.69–1.09; *p* = 0.23) [27]. In the GISCAD trial, an intermittent schedule of 5-fluorouracil plus irinotecan administered 2-months-on and 2-months-off showed equivalent progression-free survival (PFS) (HR, 1.03; 95% CI, 0.81–1.29) and OS (HR, 0.88; 95% CI, 0.69–1.14) compared to treatment continuation [28]. Conversely, the MRC COIN trial failed to meet the non-inferiority of drug holidays after 5-fluorouracil plus oxaliplatin (OS: HR, 1.08; 95% CI, 0.97–1.21 and PFS: HR, 1.05; 95% CI, 0.95–1.17) [29]. Survival outcomes of maintenance treatment were examined in the OPTIMOX1 study. FOLFOX followed by 5-fluorouracil produced similar efficacy compared to continuous FOLFOX (PFS: HR, 1.06; 95% CI, 0.89–1.20; *p* = 0.47 and OS: HR, 0.93; 95% CI, 0.72–1.11; *p* = 0.49) [10].

As to chemotherapy-free intervals (CFI), the conclusions of the OPTIMOX2, CAIRO3, and AIO-027 trials reported improvement in terms of survival in favor of the maintenance strategy compared to treatment holidays [16,22,30]. However the meta-analysis of Berry et al. evaluated 11 randomized clinical trials including OPTIMOX2, CAIRO3, and AIO-027 trials and reported no clinically significant reduction of OS between intermittent and continuous strategies (HR, 1.03; 95% CI, 0.96–1.10; *p* = 0.38) [31]. Lorée et al. also showed, in a retrospective study, that patients undergoing any de-escalation had better OS than the ones continuing a full-intensity regimen [32].Likewise in our analysis patients receiving DHs or maintenance plus DHs had similar OS (*p* = 0.461, holidays; *p* = 0.080, holidays and maintenance, taking maintenance as reference), even after correction for confounding variables. 

Summing up, although treatment maintenance showed improved PFS over chemotherapy-free intervals, noteworthy treatment holidays improved QoL in some trial, as reported in CAIRO-3 [22]. Similarly, MRC COIN demonstrated that delivering systemic chemotherapy in first-line followed by a treatment break improved social (OR, 0.82; 95% CI, 0.70–0.96; p = 0.016) and role functioning (OR, 0.82; 95% CI, 0.70–0.96; p = 0.015) compared to maintenance therapy [29]. A key difference of our study compared to the aforementioned trials is that the latter were conceived with rigid protocols imposing (since the time of randomization) the type of de-escalation after a pre-established number of cycles and did not allow adaptation of the execution on the basis of the type of response. None of the prospective studies allowed both maintenance and intermittent therapy, whereas our series also included patients treated with a sequence of intermittent and maintenance chemotherapy. Of note, we reported similar survival outcomes for both patients treated with maintenance only and with maintenance followed by a break.

Since losing the chance to access a second-line treatment could be considered a risky collaterality of de-escalation strategies, we also analyzed this aspect. In our study, the whole population attrition rate was 27%, consistent with previous literature data [33]. The attrition rate between first- and second-line treatment was 22.80%, 20.61%, and 19.64% for maintenance, a break, or both. This is reassuring, considering that attrition plays a key role in mCRC and it can hinder the benefits in terms of OS of a sequential treatment strategy. In our study, 25% of patients undergoing holidays from toxicities did not receive second-line treatment compared to 19% of patients performing holidays by the physician’s choice. This was not the case for the “early discontinuators” (mostly patients whose disease progression occurred within three months from the beginning first-line treatment). Indeed, the attrition rate was 49% in this group.

We speculated that better OS in de-escalation strategies (over continuous) could be driven by a selection bias. If physicians had offered de-escalation only to the best responders to first-line treatment or to patients with a lower burden of disease, then we could perform further analysis comparing the same treatment strategies only in the good-prognosis population. Patients with a score of ≥3 (worst prognosis) were excluded and no differences were observed amongst good prognosis patients receiving maintenance, a break, or both, while a worse outcome was seen for continuous treatment. Anyway, even if not statistically significant, in this subgroup, treatment holidays could be considered one of the main options to be offered, as it demonstrates a numerically greater advantage (HR, 0.75; 95% CI, 0.53–1.06) in terms of OS.

We are aware that this was a retrospective analysis conducted in two centers of a limited geographic area and that the study did not provide data on QoL, that could have even more supported the advantages of DHs; finally, the reasons for starting maintenance treatment are lacking. Nevertheless, some precautions were taken by the authors to overcome the weaknesses of the study: first, metastatic patients receiving metastasectomy or loco-regional treatment during first-line therapy were excluded and second, patients were stratified based on the classical groups taken into consideration in clinical trials as well as on further groups to better stratify population. In sum, even though this study embodies both weaknesses and strengths of a “real world” analysis, compared to previous data from prospective trials, it is, to the best of our knowledge, the first study evaluating the OS impact and attrition rates of different de-potentiating strategies in a “real world” setting of mCRCs. Third, a little innovation of this study was the attempt to produce a score created in order to better select patients who could safely undergo de-escalation (both maintenance and, more innovatively, DHs). Finally, these results should be considered as “hypothesis generating” in order to investigate DHs in larger future studies.

## 4. Materials and Methods

### 4.1. Study Design

This was a bi-centric, observational, retrospective, cohort study that examined data on 890 consecutive metastatic colorectal cancer (mCRC) patients who underwent first-line chemotherapy. A cohort of 608 patients selected according to inclusion and exclusion criteria entered in the final analysis. The study aimed to evaluate the impact of different treatment strategies, after induction chemotherapy, in terms of overall survival (OS). Moreover, we analyzed attrition rates according to treatment algorithms and the impact of these strategies in a population selected for good prognosis. The study was conducted in accordance with the Declaration of Helsinki, and the protocol was approved by the departmental review boards and by the Ethic Committee (Parere CEUR-2019-Os-030, Regional Ethics Committee of Friuli Venezia Giulia, Italy).

### 4.2. Patient Population

We collected the clinical data of 608 consecutive patients. All patients had confirmed histological diagnosis of mCRC and provided consent to the use of clinical data, rendered anonymous, for purposes of clinical research, epidemiology, training, and study of diseases. The cohort consisted of consecutive patients treated at the Oncology Department of the University Hospital of Udine and the Medical Oncology and Cancer Prevention Unit of the CRO National Cancer Institute of Aviano (Italy) from 1 January 2005 and 15 March 2017. Data were retrieved from an electronic and paper-based medical chart review according to strict privacy standards.

Main inclusion criteria were: age ≥18, histologically confirmed diagnosis of mCRC, having been treated with at least one line of chemotherapy. Data concerning age, sidedness, resection of primary tumor, date of metastatic disease diagnosis, pattern of metastasis, number of metastatic sites, molecular profile (KRAS, NRAS, BRAF, and all-RAS mutational status), date and type (single agent, single agent plus biologic, doublet, doublet plus biologic, triplet, triplet plus biologic, or other) of first-line chemotherapy, treatment strategy after induction therapy, metastasectomies and loco-regional therapies, and reasons for a treatment break and reinduction were collected. Finally, 608 patients were fully eligible (see CONSORTdiagram) and patients receiving metastasectomies and loco-regional therapies during first-line were excluded.

### 4.3. Definition of Drug Holiday and Treatment Strategies

Drug holidays, were defined as a treatment break of ≥56 consecutive days free from any oncology treatment after a first-line “induction” therapy. This number of minimal days was defined on the basis of the previous study by Labianca et al. [28] This period of days was considered more in line with clinical practice than other definitions of a break [32]. Maintenance chemotherapy was defined as the omission of one or more agents (i.e., oxaliplatin/irinotecan) in patients that had previously received at least one cycle of combination treatment (doublet or triplet). Treatment after induction chemotherapy was categorized into DHs, maintenance, or both combined (maintenance first, DHs later) and compared with continuous full-intensity first-line therapy. Finally, to overcome potential selection bias, a fifth group, including patients receiving less than 3 months of induction therapy (early discontinuation) were individuated.

### 4.4. Sample Size Calculation

The sample size was estimated in order to obtain a good performance of the statistical model for the association between patient and tumor characteristics with outcome measures in the multivariate analysis. The aim of the sampling was the achievement of a good “goodness of fit” for the regression model according to Peduzzi et al. [34] They showed that for number of events per variable (EPV) values of 10 or more, no major problems occurred. In particular, 20–50 events per variable (EPV) would remove the need for shrinkage of estimated regression coefficients in prespecified models.

Therefore, according this evidence, considering 50 EPV and a final model with a maximum of nine variables, it would be necessary to have 450 events. By predicting that 80% of patients will have had an unfavorable outcome at the time of the analysis (estimating a median follow-up of about 6 years, in temporal terms), and considering a final model with nine variables, it would be necessary to enroll at least 563 patients to have 450 events. This number of patients is compatible with the epidemiological data relating to these centers within the set time limits. Therefore, we could define an accurate estimation for the multivariate model.

### 4.5. Statistical Analysis

Patients’ clinical and pathological characteristics were summarized through descriptive analysis. Categorical variables were described with frequency distribution, whereas continuous variables were reported by median and range. Differences across groups were compared with the chi-square test for categorical variables. Patients alive at the time of last follow-up were censored.

For overall survival analyses, time at risk was calculated from the date of metastatic disease diagnosis to the date of the event of interest—death or last follow-up. For univariate survival analysis, OS probabilities were estimated using the Kaplan–Meier method and compared by log-rank test.

A Cox proportional-hazards regression model, also including potential confounders (e.g., age, biological profile, and sidedness) was used to calculate hazard ratios (HRs) of death, with the corresponding 95% confidence intervals (CIs), among different subgroups of patients identified by type of treatment execution. In order to define the impact of treatment executions in patients with a good prognosis, a prognostic score model based on multivariate analysis was developed. To identify a threshold to discriminate patients with good and poor prognosis, a receiving operator curve (ROC) analysis was performed. Associations between variables were explored in the whole cohort by using statistical tests (chi-square, Wilcoxon rank-sum test, or Kruskal–Wallis test), as appropriate. 

Overall survival (OS) was defined as the time between treatment start and death from any cause. Attrition rate was defined as the proportion of patients who started therapy but were not further treated at the time of disease progression due to progression itself or death, toxicity, or patient or physician decision; patients lost at follow-up were excluded from this analysis. A two-sided *p* < 0.05 was considered statistically significant. 

The data analysis was generated using STATA (StataCorp. (2015) Stata Statistical Software: Release 14.2. StataCorp LP, College Station, TX, USA).

## 5. Conclusions

In conclusion, our study supports the use of maintenance and DHs in a real-world cohort, ultimately demonstrating that physicians are able to carefully identify patients who may receive de-escalation without compromising their clinical outcomes.

## Figures and Tables

**Figure 1 cancers-13-03504-f001:**
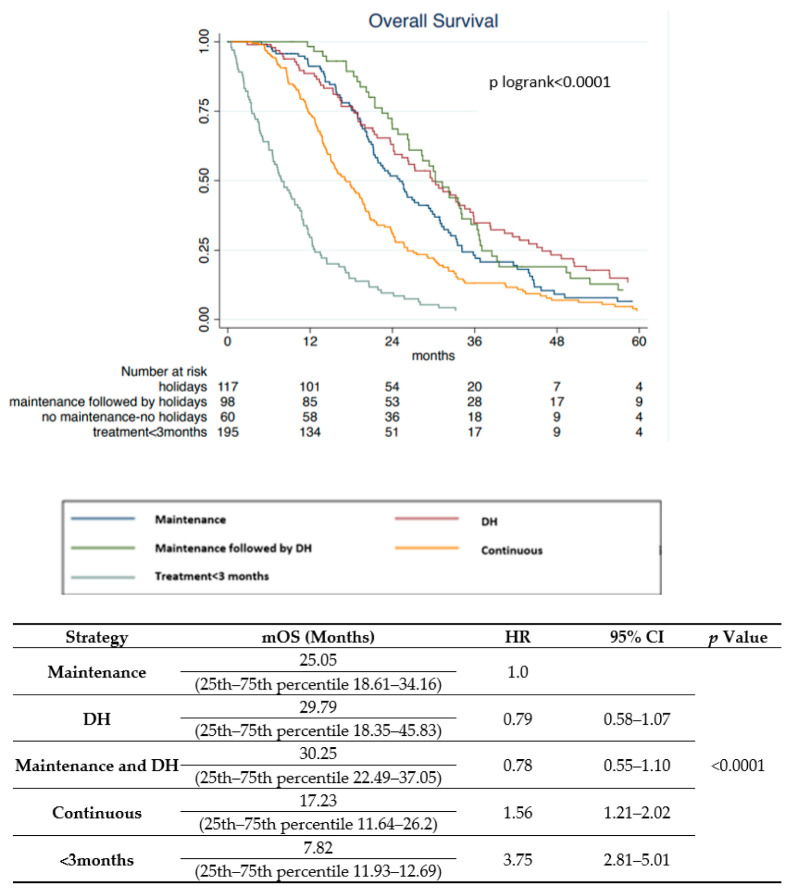
Kaplan–Meier survival curves according to strategy.

**Figure 2 cancers-13-03504-f002:**
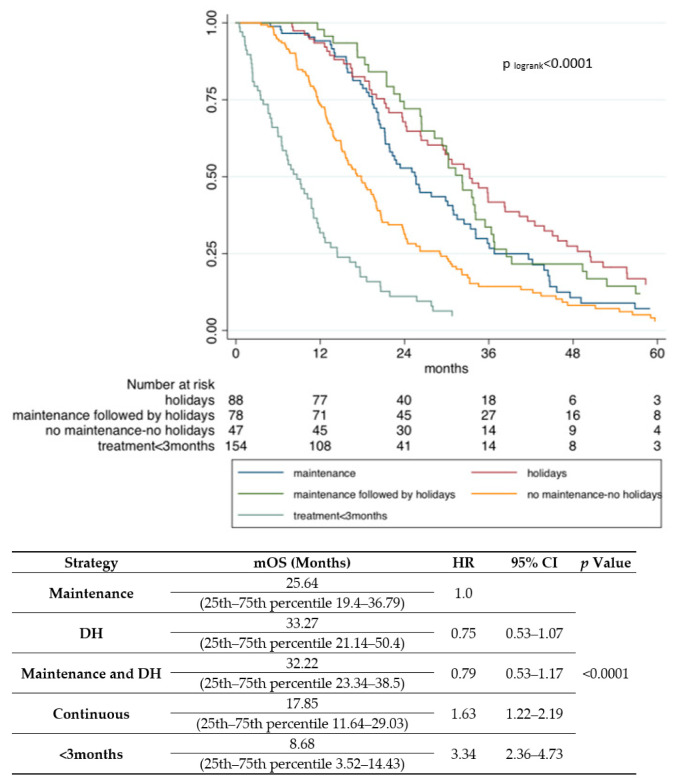
Kaplan–Meier survival curves according to strategy in patients with score <3.

**Figure 3 cancers-13-03504-f003:**
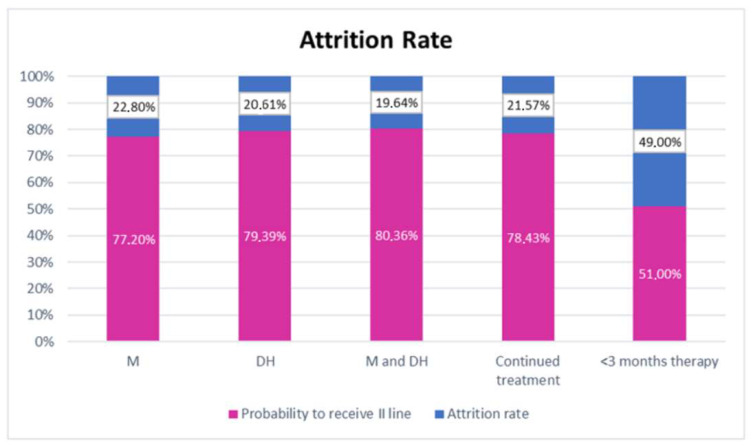
Attrition rate and probability of receiving II line treatment. M: maintenance; DH: drug holidays; M and DH: maintenance and drug holidays.

**Table 1 cancers-13-03504-t001:** Demographic and clinical characteristics of the patients at baseline. Frequencies of variables are reported by absolute number and by percentage value.

Characteristic	N Total (608 Patients)	Frequency (%)
Sex:		
Male	383	63.0%
Female	225	37.0%
Age:		
<70	390	64.14%
>70	218	35.86%
ECOG performance status:		
0	51	8.39%
1	463	76.15%
2	48	7.89%
Missing	46	7.57%
Location of primary tumor:		
Right	176	28.95%
Left	245	40.30%
Rectum	181	29.77%
Missing	6	0.99%
Pathological T stage (TNM):		
pT1	7	1.15%
pT2	31	5.10%
pT3	281	46.22%
PT4	117	19.24%
Missing	172	28.30%
Pathological N stage (TNM):		
N0	107	17.60%
N1	142	23.36%
N2	172	28.95%
N3	55	9.05%
Missing	132	21.71%
Tumor Grading:		
G1–2	244	40.13%
G3–4	156	26.66%
Missing	208	34.21%
Resection of primary tumor:		
Yes	443	72.86%
No	161	26.48%
Missing	4	0.66%
Adjuvant chemotherapy:		
Yes	87	14.31%
No	362	26.48%
N.A.	159	26.15%
Neoadjuvant chemotherapy and RT:		
Yes	76	12.50%
No	509	83.72%
Missing	23	3.78%
Metastasectomy before first-line chemotherapy:		
Yes	128	21.05%
No	478	78.62%
Missing	2	0.33%
Number of metastatic sites:		
1	352	57.89%
>1	240	39.47%
Missing	16	2.63%
Metastatic sites:		
Liver	204	33.55%
Lung	128	21.05%
Lymph nodes	102	16.78%
Peritoneum	123	20.23%
Bone	12	1.97%
CNS	6	0.99%
Missing	33	5.43%
Histotype:		
Mucinous	72	11.84%
Not mucinous	324	53.29%
Missing	212	34.87%
First-line chemotherapy:		
Single agent	72	11.84%
Doublet	191	31.41%
Doublet + biologic agent	270	44.41%
Triplet + biologic agent	57	9.38%
Other	18	2.96%
Maintenance therapy or drug holiday within first-line:		
Maintenance	117	19.24%
Holiday	98	16.12%
Maintenance and holiday	60	9.87%
Continuous treatment	195	32.07%
<3 months of chemotherapy	102	16.78%
Missing	36	5.92%
Molecular biology status:		
BRAF mut.	45	7.40%
BRAF unknown	131	21.55%
KRAS mut.	237	38.98%
KRAS unknown	93	15.30%
NRAS mut.	16	2.63%
NRAS unknown	192	31.58%
All-RAS mut.	295	48.5%
All-RAS unknown	62	10.2%
Second-line chemotherapy:		
Yes	422	69.41%
No	161	26.48%
Missing	25	4.11%
Drug Holiday:		
Yes	156	26.7%
No	421	69.2%
Missing	31	5.1%
Motivation for DH:		
Patient’s request	17	10.9%
Physician-patient shared choice	103	66.0%
Unacceptable toxicity	29	18.6%
Missing	7	4.5%

**Table 2 cancers-13-03504-t002:** Univariate and multivariate analyses for OS. Significant associations are written in bold.

	Univariate Analysis			Multivariate Analysis		
Variables	HR	*p*	95% CI	HR	*p*	95% CI
Location of primary tumor:						
Right	1.00					
Left	**0.66**	**<0.001**	**0.54–0.82**	**0.59**	**<0.001**	**0.45–0.79**
Rectum	**0.64**	**<0.001**	**0.51–0.81**	**0.56**	**<0.001**	**0.41–0.77**
Resection of primary tumor:						
No	1.00					
Yes	**0.47**	**<0.001**	**0.39–0.58**	1.37	0.17	0.87–2.17
Grading:						
G1–2	1.00					
G3–4	1.07	0.552	0.86–1.33			
Nodes:						
0	1.00					
1	1.16	0.304	0.87–1.55	1.04	0.784	0.75–1.46
2	**1.56**	**0.001**	**1.19–2.05**	**1.57**	**0.005**	**1.14–2.17**
3	**2.31**	**<0.001**	**1.61–3.30**	1.52	0.129	0.88–2.63
Adjuvant chemotherapy:						
No	1.00					
Yes	1.02	0.888	0.78–1.34			
All-RAS:						
wt	1.00					
mut	**1.38**	**0.001**	**1.14–1.66**	1.07	0.572	0.84–1.36
ECOG performance status:						
0	1.00					
1	**1.99**	**<0.001**	**1.39–2.85**	**2.48**	**<0.001**	**1.67–3.76**
2	**3.44**	**<0.001**	**2.18–5.44**	1.67	0.083	0.93–3.00
Number of metastatic sites:						
<1	1.00					
>1	**1.40**	**<0.001**	**1.16–1.67**	1.19	0.155	0.93–1.52
Metastasectomy before first-line:						
No	1.00					
Yes	**0.73**	**0.005**	**0.59–0.91**	0.86	0.291	0.66–1.13
Treatment strategy after induction:						
Maintenance	1.00	-	-	1.00	-	-
Holiday	0.76	0.12	0.59–1.07	0.94	0.758	0.66–1.35
Maintenance and holiday	0.78	0.137	0.55–1.10	0.70	0.108	0.46–1.08
Continuous treatment	**1.57**	**0.001**	**1.22–2.02**	**1.68**	**0.002**	**1.22–2.32**
<3 months of chemotherapy	**3.75**	**<0.001**	**2.81–5.01**	**4.89**	**<0.001**	**3.33–7.19**

**Table 3 cancers-13-03504-t003:** Multivariate model for prognostic factors and Score construction. Significant associations are written in bold.

	Multivariate Analysis			Score (Points)
Variables	HR	*p*	95% CI	
Right location of primary tumor	**1.57**	**<0.001**	**1.25–1.97**	**1**
All-RAS mutated	1.26	0.189	0.89–1.78	
ECOG performance status:				
1	**1.81**	**0.010**	**1.15–2.84**	**1**
2	**2.58**	**0.001**	**1.48–4.51**	**2**
Metastatic sites > 1	1.17	0.179	0.93–1.46	
Metastasectomy	0.95	0.703	0.73–1.24	
Primary tumor unresected	**1.72**	**<0.001**	**1.33–2.22**	**1**

## Data Availability

Original data used for this study are available upon request.

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
