# Peer review of "Drug Holidays and Overall Survival of Patients with Metastatic Colorectal Cancer"

_cancers, 2021, doi:10.3390/cancers13143504_

Round 1
Reviewer 1 Report
In this manuscript, the authors evaluate the impact of de-escalation treatment on OS in 608 mCRC patients. In addition, the authors study a prognostic scoring model and the attrition rates in different groups of treatment. Interestingly, authors show a benefit in terms of OS by employing this type of de-escalation strategies. The manuscript is well organized and the methods and statistical evaluation of data are explained in detail and are clear. However, a few points need to be considered:
1) In the line 143, page 3, the authors define the attrition rate, but they also define it in the line 147, page 4. Please, could the authors clarify the definition of attrition rate?
2) In the page 8, figure 1, the bottom box containing the legend, in the green legend the word “followed” is not spelled correctly.
3)In “Results”, section b, from line 202 to the end of the paragraph the authors report that score > 3 was associated with worse OS. Please, could the authors clarify here, as they indicated in supplementary figure 1, that it is a score > or = 3?
4) In “Results”, section c, line 216 the authors indicate “an association between maintenance therapy, DH or both, and the chance of receiving a chemotherapy re-induction (p=0.012)”. Please, could the authors clarify what the re-induction therapy is and what happens in this case with the subgroup of patients who had continuous combination chemotherapy (attribution rate for continuous chemotherapy are very similar to maintenance therapy, DH and both).
5) Have the authors any result about the impact of DH on PFS? In my opinion this would also be interesting from a clinical point of view.
Author Response
POINT TO POINT ANSWERS TO THE REVIEWERS
TITLE: Drug holidays and overall survival of patients with metastatic colorectal cancer
#Reviewer1
In this manuscript, the authors evaluate the impact of de-escalation treatment on OS in 608 mCRC patients. In addition, the authors study a prognostic scoring model and the attrition rates in different groups of treatment. Interestingly, authors show a benefit in terms of OS by employing this type of de-escalation strategies. The manuscript is well organized and the methods and statistical evaluation of data are explained in detail and are clear. However, a few points need to be considered:
1-In the line 143, page 3, the authors define the attrition rate, but they also define it in the line 147, page 4. Please, could the authors clarify the definition of attrition rate?
Thank you for the observation. We consider appropriate the following definition of attrition rate: “Attrition rate was defined as the proportion of patients who started therapy but were not further treated at the time of disease progression due to progression itself or death, toxicity, patient or physician decision”. The definition is now reported at line 332, and we deleted the other incorrect information.
2-In the page 8, figure 1, the bottom box containing the legend, in the green legend the word “followed” is not spelled correctly.
Thank you for the remark. We modified the text accordingly.
3-In “Results”, section b, from line 202 to the end of the paragraph the authors report that score > 3 was associated with worse OS. Please, could the authors clarify here, as they indicated in supplementary figure 1, that it is a score > or = 3?
Thank you for the correction. As reported in supplementary figure 1, the ROC curve was performed based on the subdivision <3 points or =/> of 3 points. We have modified the text accordingly, at line 133.
4-In “Results”, section c, line 216 the authors indicate “an association between maintenance therapy, DH or both, and the chance of receiving a chemotherapy re-induction (p=0.012)”. Please, could the authors clarify what the re-induction therapy is and what happens in this case with the subgroup of patients who had continuous combination chemotherapy (attribution rate for continuous chemotherapy are very similar to maintenance therapy, DH and both).
Thank you for the comment. In our series patients for whom a de-potentiation strategy is put into practice (maintenance therapy, DH or both) are good candidates for re-inducing the initial full intensity treatment at time of progression of disease (i.e FOLFOX+bevacizumab induction-> Fluorouracil+bevacizumab de-potentiation or DH-> progression: re-induction with FOLFOX+bevacizumab). This is a common strategy aimed at differing the time at which a second line treatment would be started and is supported by results from randomized phase 3 trials such as TRIBE2 trial, CAIRO3 trial and others.
In the specific case of the sentence reported at line 216 of the results section, the formulation of the phrase has been modified upon your suggestion. We actually studied the association between DH, maintenance or both and re-induction (continuous treatment was not part of the calculation). Our conclusion is therefore that there is a significative association between each of these strategies and re-induction (p=0.012). Accordingly, we have rephrased: “We looked for an association between de-potentiation strategies (DH, maintenance or both) and re-induction and we found it statistically significant (p=0.012). The strongest association was between maintenance plus DH and re-induction, followed by maintenance and then by DH alone” (lines 147 to 150 of Results section).
Finally, in the opinion of the authors, the similar attrition rates between de-potentiated strategies (maintenance, DH or both) and continuous treatment is reassuring because it suggests that “de-potentiating” is not the cause for enhancing clinical deterioration. In other words, choice in favor of de-potentiation does not cause fatal progression of disease during “milder” treatment. The same could not be stated for patients with rapid progression who deteriorate their clinical condition rapidly and do not reach the possibility of a second line treatment.
Summing up, both patients on continuous and de-potentiated strategies have the same chance of receiving a second line treatment.
5- Have the authors any result about the impact of DH on PFS? In my opinion this would also be interesting from a clinical point of view.
This is very interesting point. Unfortunately, we do not dispose of PFS but it would be of great interest to study also the impact of DH on PFS in a future evolution of this study.
Reviewer 2 Report
Dear authors,
Thank you for great manuscript, I read it with great interest. Only few minor remarks.
How was right, left colon and rectum described? Have you excluded the transverse colon?
Otherwise the discussion is very scientific.
Author Response
POINT TO POINT ANSWERS TO THE REVIEWERS
TITLE: Drug holidays and overall survival of patients with metastatic colorectal cancer
Dear authors, thank you for great manuscript, I read it with great interest. Only few minor remarks.
- How was right, left colon and rectum described? Have you excluded the transverse colon?
Thank you for this very interesting question. There is huge debate on the prognostic and predictive role of sidedness in metastatic colon cancer. If, on the one hand, we know quite well the biologic distinction between left and right colon cancer, the role of transverse cancer remains a more debated issue. For some researcher transverse colon cancer is a collection of distinct biologic types that represents a continuous of molecular changes and biologic features that embody the characteristics more of right or left cancer, dependent on the proximity to one side or the other. Therefore, in our study, we categorized transverse colon cancer as right or left considering if it was closer to one extremity or the other. Rectal cancer refers instead to the portion of large intestine that is located within 12 cm from the anus, corresponding to extra-peritoneal rectum.
Otherwise the discussion is very scientific.
Reviewer 3 Report
The topic of work and analysis are very important and translate into clinical practice. However, the work should be enriched with data from other centers.
The work is not multicenter, and only includes data from 2 centers both from North Italy and from this same region: treated at the 1)Oncology Department of Uni-96 versity Hospital of Udine and 2) Medical Oncology and Cancer Prevention Unit of CRO Na-97 tional Cancer Institute of Aviano (Italy). Moreover the target research group of 608 cases is not large for a retrospective study e.g. in work from a journal with a similar IF = 5.8 like Cancer journal research group has 39,502 cases
Bisphosphonate Drug Holiday and Fracture Risk: A Population-Based Cohort Study Annette L Adams 1 , John L Adams 2 , Marsha A Raebel 3 , Beth T Tang 2 , Jennifer L Kuntz 4 , Vinutha Vijayadeva 5 , Elizabeth A McGlynn 2 , Wendolyn S Gozansky 3 Affiliations PMID: 29529334 DOI: 10.1002/jbmr.3420
The authors should include in the analysis from other centres, as in their previous cited work “Intermittent versus continuous chemotherapy in advanced colorectal cancer: a randomized 'GISCAD' trial”, in which data came from 27
A considerable number of authors 19 draws attention as for a retrospective study conducted only on data collected from patient files from 2 centers and performed statistical analysis
In table 1 n tot is 608, in the line Molecular biology the status is different n = 1071, drug holiday n = 604, Motivation for DH n = 156 - please add an additional description, e.g. * and explain why n is different. The definition of individual groups is not clear and legible, and it wonders whether the specification of so many subgroups of maintenance, maintenance with DH, continuous treatment .... does not obscure the final conclusion and implementation of the topic of the work. Perhaps it would be helpful in the reception to emphasize the comparison of patients with DH with other individual groups (instead of putting all the groups together) and to separate section in the results highlighting the results on DH.
Minor mistakes - The authors do not follow the instructions for the journal - the layout of the work should change and the results and discussion should appear before the methodology, also the affiliation numbers next to the names should be corrected
Author Response
POINT TO POINT ANSWERS TO THE REVIEWERS
TITLE: Drug holidays and overall survival of patients with metastatic colorectal cancer
The topic of work and analysis are very important and translate into clinical practice. However, the work should be enriched with data from other centers.
1)The work is not multicenter, and only includes data from 2 centers both from North Italy and from this same region: treated at the Oncology Department of Uni-96 versity Hospital of Udine and Medical Oncology and Cancer Prevention Unit of CRO Na-97 tional Cancer Institute of Aviano (Italy).
2) Moreover the target research group of 608 cases is not large for a retrospective study e.g. in work from a journal with a similar IF = 5.8 like Cancer journal research group has 39,502 cases.
Bisphosphonate Drug Holiday and Fracture Risk: A Population-Based Cohort Study Annette L Adams 1 , John L Adams 2 , Marsha A Raebel 3 , Beth T Tang 2 , Jennifer L Kuntz 4 , Vinutha Vijayadeva 5 , Elizabeth A McGlynn 2 , Wendolyn S Gozansky 3 Affiliations PMID: 29529334 DOI: 10.1002/jbmr.3420
The authors should include in the analysis from other centres, as in their previous cited work “Intermittent versus continuous chemotherapy in advanced colorectal cancer: a randomized 'GISCAD' trial”, in which data came from 27
1/2-Thank you for the comment. We have read with interest the study “Biphosphonate Drug Holiday and Fracture Risk: A population-based cohort study” by Annette L. Adams et. Al. Indeed it is a very large cohort retrospective multi-regional study including 39,502 cases. In the cited study, the availability of data from integrated regional healthcare systems has made possible to retrieve an impressive amount of data. Unfortunately, we did not dispose of such a large region-based dataset, which is a good support for collecting medical data but presupposes a very efficient “regional healthcare system level” collection of medical information. In the case of our study, we disposed of “Oncology Department level” clinical data because regional systems are, nowadays, not ready to record the vast treatment details that our study required. Since our study was conducted in two centers of the same Italian Region, we have corrected the way you have suggested. At line 253 we have deleted the adjective multicentric and converted to “bi-centric”. The reason for conducting a retrospective study in these two centers is the availability of complete clinical data that could be provided thanks to their advanced medical data archive.
2-The total of 608 patients is the result of a consistent retrospective data collection effort. Indeed, at the beginning of the study, we had a database of 1266 patients of whom: 224 were excluded because they had received only best supportive care and 152 patients presented a too short follow up. We then obtained 890 patients but 282 have been excluded because they had received a metastasectomy or thermoablation during first line treatment (exclusion criterion). The final analysis was performed on 608 patients.
Nevertheless, this remains a quite large database also considering 608 patients. Moreover, this number of patients is sufficient to guarantee the good quality of the results because, at time of trial designing, we estimated a sample size in order to obtain a good performance of the statistical model. In particular, we estimated a maximum of 9 variables in the final model and 50 EPV (event per variable), that is the highest number designated by statisticians to remove the need for shrinkage of estimated regression coefficients in pre-specified models. In our sample size we needed at least 563 patients to have 450 events. We could therefore define an accurate estimation of our multivariate model as we had 608 patients and 509 total events in our database. For more transparency, as requested, we have added a paragraph that specifically explains sample size calculation in the methods session (line 295 to 310).
We also agree that our work should be considered as “hypothesis generating” and it could be reinforced by extending to other Italian centers in the future. Further evolution could include a larger number of Oncology Departments with prospective design as it was done in the randomized clinical trial by GISCAD (a large internationally recognized Italian collaborative trialists group). We keep your suggestion and we intend to build on your remark for the next future of this project related to drug holidays in colorectal cancer.
Upon your observation, we have updated line 234 and we have edited the sentence of discussion as follows:” We are aware that this is a retrospective analysis conducted in two centers of a limited geographic area and that the study does not provide data on QoL, that could have even more supported the advantages of DHs; finally, the reasons for starting maintenance treatment lack.” and also added on line 247 “These results should finally be considered as “hypothesis generating” in order to investigate DHs in larger future studies”.
3- A considerable number of authors 19 draws attention as for a retrospective study conducted only on data collected from patient files from 2 centers and performed statistical analysis
Thank you for your observation. Although the number of authors may seem disproportionate to the work done, it actually reflects the commitment that was required for the different components of the study:
- Extraction and collection of very complex clinical data from 12 years of visits (retrieved from mixed electronic and paper records);
- Follow up contacts: a large number of telephone follow up calls
- Involvement in study design and elaboration
4-In table 1 n tot is 608, in the line Molecular biology the status is different n = 1071, drug holiday n = 604, Motivation for DH n = 156 - please add an additional description, e.g. * and explain why n is different.
Thank you for your remark but, in fact, the numbers are correctly reported. Please consider our answer below.
-the total number of patients is 608
-In the molecular biology status line: the vertical line cannot be summed (n=1071) because KRAS, NRAS, BRAF and all-RAS mutations are not all mutually exclusive. Indeed, all-RAS mutation contain KRAS and NRAS mutation. Moreover, it is generally demonstrated that in the majority of cases KRAS mutation is mutually exclusive with BRAF mutation but the same is not always true for NRAS which may co-exist. Every mutation should then be considered on its own.
-Drug Holiday: vertical sum is 608
-As to motivation of DH: the sum of the motivations (patient’s request, physician-patient shared choice, unacceptable toxicity and missing) should be 156 and not 608 because it refers only to the patients that have received DH (156)
5-The definition of individual groups is not clear and legible, and it wonders whether the specification of so many subgroups of maintenance, maintenance with DH, continuous treatment .... does not obscure the final conclusion and implementation of the topic of the work. Perhaps it would be helpful in the reception to emphasize the comparison of patients with DH with other individual groups (instead of putting all the groups together) and to separate section in the results highlighting the results on DH.
We categorized our study population into four major groups of strategies: maintenance, DH, maintenance plus DH and continuous treatment. In fact, at the moment of study design, we believed that they do not have the same clinical characteristics as well as they are probably offered to patients with different clinical and biological behaviors. Also, we felt that they could generate different outcomes. Finally, this subdivision by strategy has been possible because of our initial sample size calculation and the final large number of events that we had at our disposition.
In the opinion of the authors, it would be misleading to assimilate these three categories in one single group.
6- Minor mistakes - The authors do not follow the instructions for the journal - the layout of the work should change and the results and discussion should appear before the methodology, also the affiliation numbers next to the names should be corrected
Thank you for the correction. We have placed results and discussion before methodology and we have corrected the affiliation numbers next to the names. In case of further layout issues, we will be at complete disposition to modify as requested by the editor.
Round 2
Reviewer 3 Report
.